# Percutaneous Closure of Mitral Paravalvular Leak: Long-Term Results in a Single-Center Experience

**DOI:** 10.3390/jcm11164835

**Published:** 2022-08-18

**Authors:** Ignacio Cruz-González, Pablo Luengo-Mondéjar, Blanca Trejo-Velasco, Jean C. Núñez-García, Rocío González-Ferreiro, José C. Moreno-Samos, Mónica Fuertes-Barahona, Juan C. Rama-Merchán, Pablo Antúnez-Muiños, Sergio López-Tejero, Gilles Barreira de Sousa, Javier Rodríguez-Collado, Javier Martín-Moreiras, Alejandro Diego-Nieto, Jesús Herrero-Garibi, Manuel Barreiro-Pérez, Elena Díaz-Peláez, Pedro L. Sánchez Fernández

**Affiliations:** 1Department of Cardiology, University Hospital of Salamanca, 37007 Salamanca, Spain; 2Instituto de Investigación Biomédica de Salamanca (IBSAL), 37007 Salamanca, Spain; 3Centro de Investigación Biomédica en Red Enfermedades Cardiovasculares (CIBERCV), 28029 Madrid, Spain; 4Department of Cardiology, University Hospital of Burgos, 09006 Burgos, Spain; 5Department of Interventional Cardiology, Leipzig Heart Institute, 04289 Leipzig, Germany; 6Department of Cardiology, University Hospital Alvaro Cunqueiro, 36213 Vigo, Spain; 7Department of Interventional Cardiology, King’s College Hospital, Dubai P.O. Box 340901, United Arab Emirates; 8Department of Cardiology, University Hospital Mútua Terrassa, 08221 Terrassa, Spain; 9Department of Cardiology, Hospital of Mérida, 06800 Mérida, Spain

**Keywords:** paravalvular leaks, mitral regurgitation, heart failure, hemolytic anemia, valvular prosthesis, percutaneous closure

## Abstract

Background: Paravalvular leak occurs in 5–17% of patients following surgical valve replacement, more often in mitral position. The prognosis without treatment is poor. Percutaneous device closure represents an alternative to repeat surgery. The objective of this work is to evaluate the medium and long-term results in the percutaneous closure of PVL in mitral prosthesis. Methods: This observational study is based on a retrospective registry including consecutive mitral PVL cases undergoing percutaneous closure at a single tertiary-care center from April 2010 to December 2020. The safety and efficacy results of the procedure, at 90 days and in the long term, were analyzed. Also, predictors of procedure failure and long-term events were identified. Results: A total of 128 consecutive mitral paravalvular leak closure procedures were included. Technical success was achieved in 115 (89.8%) procedures. The presence of multiple PVLs was the sole factor that independently predicted procedural failure. Median follow-up of our sample was 41.8 months (mean 47.7 ± 35.7 months). Underlying hemolytic anemia as the indication for PVL closure, a recent admission for decompensated HF, and lack of improvement in functional class emerged as consistent predictors of MACE and death during long-term follow-up, while lack of procedural success during the first PVL procedure and chronic kidney disease were also associated with MACE during follow-up. Conclusions: Percutaneous mitral PVL closure displayed high technical and procedural success rates, with an acceptable safety profile, in a high-risk population. Percutaneous mitral PVL closure achieved an improvement in short- and long-term functional class and a reduction of hemolysis in the vast majority of patients. In addition, long-term survival in our study was good, in particular for patients undergoing successful PVL closure procedures.

## 1. Introduction

Paravalvular leaks (PVL) are the most common cause of nonstructural prosthetic valve dysfunction after surgical valve replacement, affecting between 7–17% of mitral valve prosthesis across the different series [1,2]. Mitral PVLs are more frequent in patients with mechanical as opposed to biological prosthesis [3,4]. Other relevant predisposing factors include mitral annular calcification, a more advanced age, or underlying chronic kidney disease, amongst others [5,6,7]. Albeit most PVLs are small in size and remain asymptomatic, up to 2–5% can become clinically relevant on account of significant paravalvular insufficiency [1,2,5]. The most frequent clinical presentation is heart failure (HF) in up to 90% of patients [8,9], while hemolysis has been documented in up to 38% of patients, most frequently in association with HF [2,9].

European Society of Cardiology (ESC) and American Heart Association (AHA) guidelines still recommend surgical reintervention as the first therapeutical option in patients with significant PVLs [10,11]. Indeed, surgery has proven effective at reducing PVL severity and has been associated with improved functional class and survival in some studies [12,13,14]. However, surgical PVL closure entails high morbidity and mortality rates so that patients at higher operative risk are often rejected for surgery. Accordingly, percutaneous transcatheter PVL closure emerges as an attractive treatment alternative which should be considered based on the patient’s surgical risk and local experience [10,11,15,16,17,18]. The number of transcatheter PVL closure procedures has increased over the years, with good short-term results and a low rate of complications [15,16,17,18], but data regarding longer-term outcomes are still limited.

The objective of this study was to evaluate the medium- and long-term outcomes of patients undergoing percutaneous transcatheter mitral PVL closure procedures in one of the most extensive series to be published to date.

## 2. Materials and Methods

This observational study is based on a retrospective registry including consecutive mitral PVL cases undergoing percutaneous closure at a single tertiary-care center from April 2010 to December 2020. The safety and efficacy results of the procedure at 90 days and in the long term were analyzed. In addition, separate analyses to search for predictors of procedure failure and long-term events were performed. The study protocol complied with the Helsinki Declaration and was approved by the Ethics Committee (of Hospital Universitario de Salamanca) and all subjects provided informed consent prior to the procedure.

### 2.1. Study Population

A total of 128 consecutive mitral paravalvular leak closure procedures performed on 96 patients at our center from April 2010 to December 2020 were included. All cases were presented in a “Heart Team” session that weighed surgical versus percutaneous closure alternatives for each individual patient. Included PVLs generated either significant symptomatic mitral regurgitation or severe hemolytic anemia. All cases were presented in a “Heart team” medical session, considering percutaneous closure. Patients with PVL that affected more than one-third of the circumference of the prosthetic ring were not considered candidates for percutaneous closure.

### 2.2. Baseline and Procedural Variables

PVL was defined as the presence of a regurgitation jet by Doppler echocardiography, originating between the edge of the prosthetic valve ring and the surrounding native tissue. The severity of paravalvular regurgitation was defined according to the criteria established by the American Society of Echocardiography [19,20], and its location was described according to the classification established by Cortes et al. [21].

The diagnosis of HF was clinical, according to the Framingham criteria, and functional class was assessed according to the New York Heart Association (NYHA) classification. Natriuretic peptide markers were not analyzed as they were not available for most patients at the time of the procedure in our center. Hemolytic anemia was defined as a plasma hemoglobin level ≤ 14 g/dL in men or ≤12 g/dL in women, in conjunction with a hemolytic profile (LDH ≥ 600 U/L, haptoglobin ≤ 10 mg/dL) and/or requirement for red blood cells transfusions in the last year as a result of anemia; transfusions due to acute bleeding were not taken into account for the definition of hemolytic anemia.

Technical success was defined as successful delivery of a PVL closure device without interference with the valve prosthesis and not requiring emergency cardiac surgery [17] and procedural success as technical success and at least 1 grade of regurgitation reduction.

Periprocedural adverse events and safety events occurring during index admission were registered and included all-cause and cardiovascular death, stroke, myocardial infarction, complete atrioventricular block, air embolism, device embolization, prosthetic leaflet impingement, emergency cardiac surgery, significant pericardial effusion or cardiac tamponade, and vascular complications.

Long-term efficacy and clinical data including NYHA functional class were recorded at 90 days postprocedurally and over a longer-term follow-up, throughout the duration of the study.

### 2.3. PVL Closure Technique and Characteristics of the Procedure

Percutaneous mitral PVL closure can be performed via a transfemoral (antegrade or retrograde) or transapical approach. PVL transcatheter technique has been extensively described previously [1].

In brief, PVL are crossed by means of a 0.035″ hydrophilic guidewire (e.g., Terumo guidewire, Terumo Medical-Corporation) over a diagnostic catheter (multipurpose or Judkins right). Generally, the next step involves establishing an arteriovenous loop by snaring the guidewire in the aorta (in antegrade procedures) or in the left atrium (in retrograde procedures). Finally, a delivery sheath is advanced over the loop across the PVL, and a closure device is deployed (Figure 1). PVL closure through transapical access can be performed percutaneously or with a mini-thoracotomy and requires retrograde PVL crossing, albeit no loop formation is generally required.

### 2.4. Imaging Data

The size of the PVLs was assessed by transesophageal echocardiography (TEE) in all patients employing three-dimensional TEE to obtain 3D images, with subsequent postprocessing of full volume images with multiplanar reconstruction tools. Additionally, cases deemed to be more complex, including those undergoing repeat procedures, due to failure of the previous procedure or to the presence of multiple leaks, underwent a preprocedural synchronized cardiac CT evaluation in a 256-slice CT scan (Philips^®^, Andover, MA, USA). Procedures were carried out employing 3D TEE or CT-image fusion superimposed to fluoroscopy, at the discretion of the operator.

### 2.5. Clinical and Echocardiographic Follow-Up

Follow-up was performed in a cardiology outpatient clinic. All patients underwent a clinical and TEE assessment 90 days after the procedure. If development of a new paravalvular dehiscence was suspected during follow-up, a repeat TEE was performed.

During long-term follow-up, the following variables were collected from the patient’s clinical history: NYHA functional class, need for red blood cell transfusion after discharge, need for reintervention (new PVL closure procedure or valve surgery), admission for HF, all-cause death, cardiovascular death, and the degree of valvular regurgitation postprocedure, assessing the decrease of at least 1 degree with respect to baseline. In addition, major adverse cardiovascular events (MACE) comprising new PVL closure intervention, HF admission, and all-cause death were registered as a separate endpoint.

### 2.6. Statistical Analysis

Normally distributed data are presented as mean ± standard deviation (SD) and nonGaussian data are presented as median ± interquartile range (IQR). Normality was tested by means of the Kolmogorov–Smirnov test. Categorical data are presented as frequencies (percentages). Changes in NYHA class (ordinal data) between baseline and following PVL closure were analyzed using the Wilcoxon rank test. Univariate (variables available in Appendix A) and multivariate analyses were performed to find predictors of PVL closure procedural success using binary logistic regression analysis.

Clinical and procedural variables were tested for an association with death and MACE during follow-up using univariable Cox proportional hazard regression. For continuous variables, the hazard ratio (HR) represented the increase in hazard per unit rise in that variable. For the categorical variables, the comparison was between the presence or absence of that factor. Those factors with *p* < 0.1 on univariable analysis (variables available in Appendix A) were entered into a multivariable model. MACE was defined as a composite of death, heart failure admission, repeat procedure, or valve surgery. Survival estimates with 95% confidence intervals (CIs) were calculated using the Kaplan–Meier method with the log-rank test. All statistical analysis was performed using the STATA statistical package (version 13.1, StataCorp, College Station, TX, USA).

## 3. Results

A total of 128 consecutive percutaneous mitral PVL closure procedures performed in 96 patients were included. Mean age was 71.1 ± 8.1 years and 53% of the patients were female, Table 1. Included patients displayed an advanced functional class at baseline (NYHA III-IV/IV in 88.5%) and an increased surgical risk (mean STS Score: 5.5 ± 4.0, mean logistic EuroSCORE 20.2 ± 10.1). The majority of patients (83.3%) had a mechanical mitral prosthesis and severe preprocedural PVLs (78.9%). Overall, HF was the most common indication for PVL closure: 59% isolated HF, 35% HF with concomitant hemolytic anemia (HA).

### 3.1. Procedural Characteristics

Technical success was achieved in 115 (89.8%) procedures and procedural success in 102 (79.2%), Table 2. Planned stage procedure was performed in 16 patients (16.7%). The most frequent PVL locations were anterior or lateral (both 32.29%), followed by septal (13.54%). More than one device was implanted in 32% of procedures, and in four cases (4.2%), a combined mitral and aortic procedure was performed. The majority (97.6%) of procedures were performed through femoral vascular access, 64.8% of them employing an antegrade approach, while transapical access was only employed in three (2.4%) cases. An arteriovenous circuit was established in 111 (86.7%) procedures and the predominant implanted device used was the Amplatzer Vascular Plug III (AVP III, Abbott Vascular^®^, Chicago, IL, USA) in 110 (90.16%) procedures, Table 2.

The most frequent procedural complications were vascular complications occurring in 12 (9.4%) procedures and comprised femoral pseudoaneurysms or hematomas requiring surgical treatment or red blood cell transfusion. In addition, there were 3 (2.4%) procedural device embolizations and 6 (4.8%) cases of device interference with the prosthetic discs: all these cases could be solved percutaneously by removing the device in the same procedure. Two (1.6%) patients required urgent cardiac surgery on account of interference of the closure device with the mechanical mitral prosthesis in one case and inadvertent puncture of the aorta during transeptal puncture in the other. The majority of patients (80.5%) did not develop any procedural complication, and there were no periprocedural deaths.

As to in-hospital outcomes, 72.7% of patients did not present any complication during hospital admission postprocedure. There were five vascular complications that developed 24 h after the procedure, three patients presented a stroke (one hemorrhagic, one ischemic, and one transient ischemic attack) and there were six (6.3%) in-hospital deaths, all of which occurred more than 7 days after admission.

One (0.76%) patient required pericardiocentesis due to cardiac tamponade at 72 h, and one (0.76%) required a temporary pacemaker implant on account of a third-degree AV block.

### 3.2. Predictors of Procedural Success

Factors associated with procedure failure are shown in Table 3. The only factor independently associated with PVL closure procedure failure was the existence of multiple PVLs (*p* = 0.03). Hemolytic anemia as the indication of PVL closure displayed a trend but did not reach statistical significance (*p* = 0.054).

### 3.3. Medium- and Long-Term Outcomes

Median follow-up of our sample was 41.8 months (mean 47.7 ± 35.7 months).

An improvement in functional class as compared to baseline functional class was observed in 78.1% patients at 90 days (Wilcoxon test with *p* < 0.0001) and in 77.4% patients at last follow-up (Wilcoxon test with *p* < 0.0001 as compared to baseline functional class), Table 4. Overall, 82.1% of patients presented a functional class NYHA I-II/IV at last follow-up visit and this percentage raised to 88.7%, 91.8%, and 97.4% in patients that had survived 1, 2, and 4 years, respectively.

Mean hemoglobin and LDH levels at follow-up [10.9 (SD 2.1) g/dL and 654 (SD 640.9) U/L, respectively] were significantly higher as compared to admission values (*p* < 0.005 for both comparisons), Appendix A, and there was a significant decrease in the need for red blood cell transfusions (*p* = 0.005).

During follow-up, 55 patients (57.3%) experienced a MACE, including death in 38 patients (39.6%), need for a repeat PVL closure intervention (either surgical or percutaneous) in 14 patients (14.6%), and HF readmission in 28 patients (29.2%). A cardiovascular cause of death was identified in 67.6% cases, mainly due to advanced HF, while the cause of death was unavailable in three (2.7%) cases. In addition, no episodes of stroke were documented during follow-up. Regarding patients who required a new intervention, 11 underwent a new percutaneous closure (11.5%) and 3 underwent conventional surgery (3.1%). 

Survival rates after the first PVL closure procedure were 75% at 1 year, 64.3% at the second year, 51.3% at 4 years, and 45.2% at 5 years, as depicted by the Kaplan–Meier survival analysis in Figure 2.

Factors associated with MACE (death, new PVL closure intervention, or HF admission) during follow-up are shown in Table 5. Factors independently associated with MACE were chronic kidney disease (HR 2.16 [1.03–4.58]; *p* = 0.001), HF admission during the previous 3 months (HR 5.68 [2.33–13.81]; *p* < 0.001), lack of procedural success during the first PVL procedure (HR 0.14 [0.04–0.48]; *p* = 0.002), lack of functional class improvement over follow-up (HR 0.36 [0.17–0.80]; *p* = 0.011), and hemolytic anemia as the indication of PVL closure (HR 5.86 [2.81–12.24]; *p* < 0.001).

Factors associated with all-cause death during follow-up are shown in Table 6. The only factors independently associated with death were lack of functional class improvement at follow-up (HR 0.23 [0.09–0.63]; *p* = 0.004), HF admission during the previous 3 months (HR 7 [2.41–20.29]; *p* < 0.001), a persistent NYHA IV functional class 90 days after the PVL closure procedure (HR 4.49 [1.16–17.46]; *p* = 0.030), and hemolytic anemia as the indication of PVL closure (HR 6.74 [2.75–16.53]; *p* < 0.001).

Kaplan–Meier plots for survival and survival free from MACE for the overall sample and according to the indication for PVL closure (HF vs. AH) are shown in Figure 3 and Figure 4, respectively.

## 4. Discussion

The main findings of this single-center registry are as follows:(1)Percutaneous mitral PVL closure procedures can attain high procedural success rates, with an acceptable safety profile, considering the high-risk profile of target patients;(2)Short- and long-term clinical outcomes after mitral PVL closure are favorable, with sustained improvement in functional class in over 75% of patients during long-term follow-up and 51% survival at 4 years;(3)The presence of multiple PVLs was the sole independent predictor of procedural failure;(4)Underlying hemolytic anemia as the indication for PVL closure, recent admissions for decompensated HF, and lack of improvement in functional class emerged as consistent predictors of MACE and all-cause death during long-term follow-up.

Management of mitral PVLs either by surgery or percutaneously is technically challenging on account of the tortuous trajectory and complex anatomy of these defects, which generally develop in highly calcified mitral valves [1,2,8,22]. In addition, patients with mitral PVL often display a high surgical risk and increased morbidity and mortality rates, which remain high despite successful PVL closure [12,13,14,15,23].

Although current guidelines recommend surgery as the first therapeutic alternative, percutaneous PVL closure is being increasingly adopted, given its better safety profile [13,14,15,16]. Currently, there are no randomized controlled trials comparing the efficacy of percutaneous closure versus surgical repair, but data from retrospective single-center registries point towards greater technical success with surgery, at the cost of increased in-hospital morbidity and mortality [23,24]. Accordingly, further evidence to support clinical decisions in the management of mitral PVL patients is needed, including refinement of patient selection criteria and identification of predictors of procedural success. Moreover, data on longer-term outcomes following percutaneous PVL closure as compared to surgery remain currently limited and should be further investigated.

In the current study, we reported high technical and procedural success rates at 89.9% and 73.2%, respectively, and an improvement in NYHA class of 77.9% at 90 days and 77.4% over long-term follow-up. In addition, we observed a substantial improvement in hemoglobin (*p* < 0.001) and LDH levels (*p* = 0.006), as well as a lower need for transfusions during follow-up (*p* = 0.005). Our findings confirm those of previous series, which have described technical success rates ranging between 77% and 91% of cases, with a reduction in at least one NYHA functional class grade in 66% to 77% of patients [15,16,17,18,22] and, thus, reinforce the value of percutaneous mitral PVL closure as an effective therapeutical alternative.

Regarding procedural safety, 80.5% of cases in our study were free from procedural-related complications despite the high-risk clinical profile of the included patients. Of importance, the majority of complications could be solved without the need for surgery, including six cases of device embolization or interference with the prosthetic valve and there were no periprocedural deaths. In-hospital mortality occurred in 6.3% of patients in our series. This value is slightly higher than the 3.9% to 5.6% reported by previous studies, driven by events in patients with end-stage HF and a high load of associated comorbidities, many of which underwent PVL closure during an admission for HF decompensation. Indeed, studies reporting outcomes following urgent PVL closure procedures have described in-hospital mortality rates as high as 6.8% for urgent procedures and up to 50% for emergent procedures [18]. A further factor that could justify our findings is the higher surgical risk of our sample as compared to other prior series, as depicted by greater Log. EuroSCORE I values: 20.6% vs. 17.5% in the HOLE registry [17]. Moreover, in-hospital rates in our study were substantially lower than reported following surgical PVL closure, which lay between 8.8% and 11.5% [25,26].

Our trial adds to current knowledge as it provides one of the longest follow-up periods following percutaneous mitral PVL closure in the scientific literature, with a median FU of 41 months. Overall survival was 75% at 1 year, 65.3% at 2 years, 51.3% at 4 years, and 45.2% at 5 years. Survival free from MACE was 42.7% at last follow-up.

To date, there are only three studies reporting long-term outcomes ≥2 years after percutaneous mitral PVL closure procedures [24,27,28]. In 2017, Alkhouli et al. reported a long-term survival rate of 50.3% following percutaneous mitral PVL closure in 195 patients, with a mean follow-up period of 3.7 ± 2.7 years [24]. More recently, Onorato et al. described a 91.3% technical success rate employing the dedicated Occlutech PVL device (Occlutech, GmbH, Jena, Germany) in a single-center registry including 93 mitral PVL and 44 aortic PVL procedures [27]. Overall mortality rate at 2 years was 8.3% in this study. Lastly, a single-center registry by Perl et al. [28] including 100 consecutive transcatheter PVL closure procedures (74 mitral, 26 aortic) reported similarly high procedural success rates (88%), albeit with a substantially higher incidence of mortality at 1-year (15.6%) and 5-year follow-up (27.2%). Although long-term mortality and MACE rates in our study were considerable, particularly beyond 1-year follow-up, they were similar to those reported by Alkhouli et al. [24] and Perl et al. [28]. Regarding the lower mortality rates observed in the Occlutech registry [27], there are several factors that limit an accurate comparison with the results of our study: first, patients included in the Occlutech trial were younger (mean age 66.7 vs. 71.1 years) and presented a lower prevalence of underlying comorbidities such as chronic kidney disease (15.3% vs. 30.2%) and COPD (5.6% vs. 9.4%), amongst others. Furthermore, the Occlutech trial reported outcomes for the whole sample comprising mitral and aortic PVL closure procedures, while a mitral versus aortic PVL location has been previously associated with a higher rate of events during follow-up [17,18].

Overall, we presume that the long-term outcomes observed in our study can be at least partially justified by the high-risk baseline profile and poor survival rates of patients undergoing mitral valve replacement, as previously reported [29,30]. Of note, surgical as opposed to percutaneous PVL closure did not improve overall survival when adjusted by baseline patient’s characteristics, which depicted a substantially higher risk profile among patients assigned for percutaneous therapy [24]. Although prior surgical series have reported survival rates as high as 39% on long-term follow-up 12 years after the procedure [27], these results cannot be compared directly with the findings of our study, since surgery was performed in a younger and lower risk patient population.

Another relevant contribution of the current study is the identification of procedural and long-term outcomes. In our sample, the existence of multiple PVLs was the sole independent predictor of procedural failure. This finding is relevant, since multiple PVL are not infrequent, appearing in up to one-fifth of patients in our sample and evidence on the management of such cases is scarce, based solely on case reports and cases series. Our results underline the importance of an accurate baseline imaging assessment in patients with PVL and identification of multiple PVLs must be taken into account when weighing surgical versus percutaneous therapeutical options in a given patient. On the other hand, hemolytic anemia as the indication for PVL closure as opposed to HF displayed a trend towards more frequent procedural failure, but this association was not statistically significant. An explanation for this linkage is that hemolysis can result from eccentric or small-sized PVLs, and closure of such defects without interfering with the surgical prosthesis can be technically challenging. Certainly, reduced clinical success rates in patients undergoing PVL closure for hemolysis versus HF (66% vs. 100%) have been previously reported [31]. Of note, our study did not identify the use of a specific closure device over another as a predictor of procedural outcomes. These results are at variance with prior findings, which have reported better procedural outcomes with the use of the AVP-III as opposed to other nondedicated devices. However, the fact that an AVP-III device was used in over 90% of procedures in our study could have hindered identification of underlying differences.

In addition, the current study identified several predictors of death and the combined MACE endpoint during long-term follow-up. Overall, the presence of hemolytic anemia as the indication of PVL closure, HF admission during the previous 3 months, and lack of improvement in functional class during follow-up emerged as independent predictors for both all-cause death and MACE on long-term follow-up. In addition, baseline chronic kidney disease and lack of procedural success during the first PVL procedure were also predictors of MACE during follow-up. These findings align with the results of prior studies describing an association between residual paravalvular regurgitation and clinical events (surgical reintervention and functional class) following PVL closure [32]. In addition, a prior single-center registry reported excess mortality rates in patients with more than mild residual PVL [23], but this association could not be confirmed in our study. Notwithstanding, the percentage of patients displaying a decrease of at least 1 degree in paravalvular regurgitation increased from 79.2% across the whole sample to 86.1%, 90% and 87.5% of patients surviving 1, 2, and 4 years, respectively, which reflects the importance of attaining optimal procedural results on long-term patient management. 

Altogether, the results of this study reinforce the value of percutaneous closure of mitral PVL as a reliable and safe therapeutic option for patients with symptomatic PVL and high surgical risk. Our series provides relevant evidence by depicting an improvement in symptoms and good short- and long-term survival rates following percutaneous closure of PVL. Furthermore, it identifies predictors of procedural and long-term outcomes, which can be of value to support clinical decision-making.

## 5. Limitations

This is a nonrandomized, retrospective, observational study. The results of our study are from a single-center experience and with a relatively small number of patients included, although it is the largest reported mitral PVL closure series with long-term follow-up. Criteria for patient triage to percutaneous PVL closure versus redo surgery was at the physicians’ discretion and may have varied over time, introducing potential bias. The clinical and echocardiographic results were self-reported and there was no independent adjudication.

## 6. Conclusions

In this single-center retrospective registry, percutaneous mitral PVL closure displayed high technical and procedural success rates, with an acceptable safety profile, in a high-risk population. Percutaneous mitral PVL closure achieved an improvement in short- and long-term functional class and a reduction of hemolysis in the vast majority of patients. In addition, long-term survival in our study was good, in particular for patients undergoing successful PVL closure procedures. The presence of multiple PVLs was the sole factor that independently predicted procedural failure. In addition, underlying hemolytic anemia as the indication for PVL closure, a recent admission for decompensated HF, and lack of improvement in functional class emerged as consistent predictors of MACE and death during long-term follow-up, while lack of procedural success during the first PVL procedure and chronic kidney disease were also associated with MACE during follow-up.

## Figures and Tables

**Figure 1 jcm-11-04835-f001:**
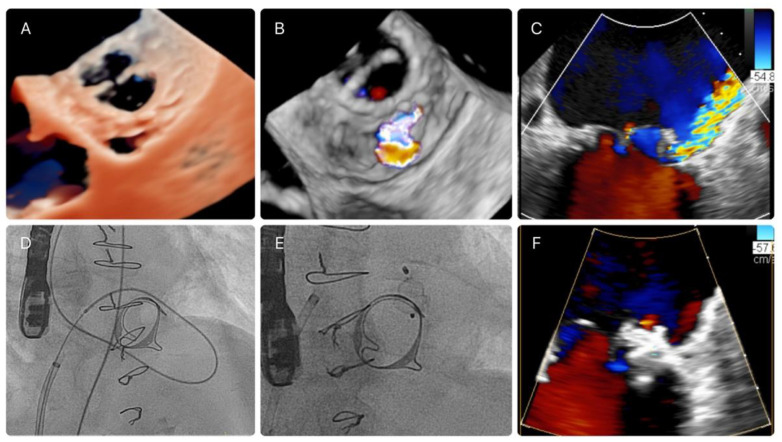
(**A**,**B**): mitral PVL by TEE with true-lumen technology and 3D-color TEE. (**C**): severe mitral regurgitation (septal PVL). (**D**): AV loop in a bioprosthesis. (**E**): AVP-III device. (**F**): nonsignificant mitral regurgitation after PVL closure (Doppler color TEE).

**Figure 2 jcm-11-04835-f002:**
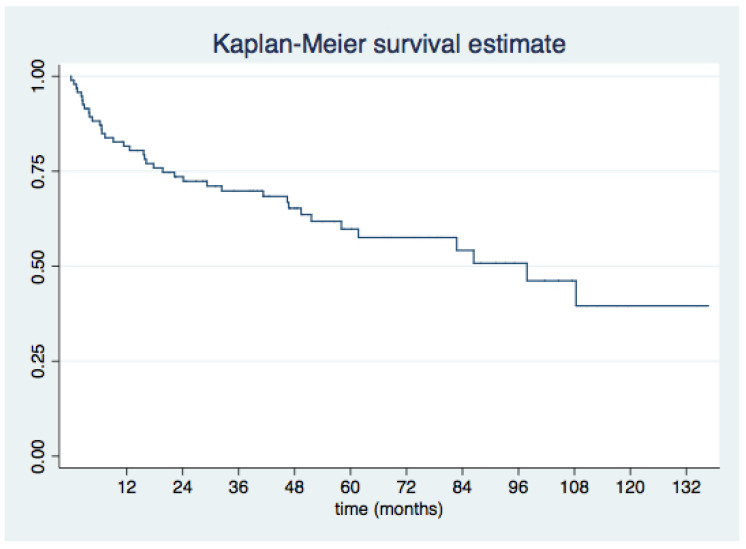
Overall survival after a first percutaneous mitral PVL closure procedure.

**Figure 3 jcm-11-04835-f003:**
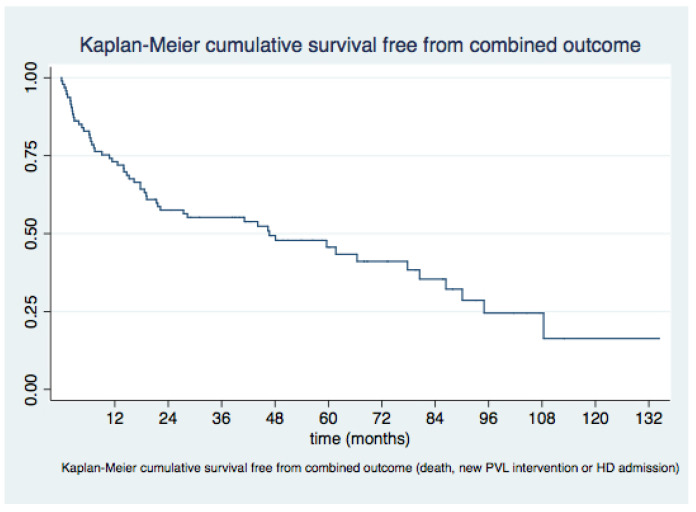
Survival free from major adverse cardiovascular events (MACE) for the overall sample.

**Figure 4 jcm-11-04835-f004:**
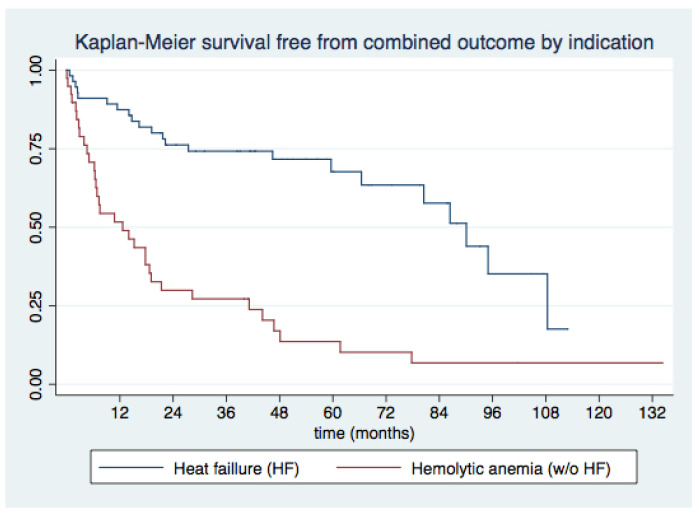
Survival free from major adverse cardiovascular events (MACE) according to indication for PVL closure.

**Table 1 jcm-11-04835-t001:** Baseline characteristics.

Patients (*n*)	96
Age (years)	71.1 ± 8.1
Sex, men (*n*, %)	46.9%
Patients with 1 prosthetic valve (*n*, %)	56 (58.3%)
Patients with 2 prosthetics valves (*n*, %)	40 (41.7%)
Patients with ≥2 prior sternotomies (*n*, %)	27 (28.1%)
Time from last surgery to procedure (months)	65.3 (IQ 126.3)
Mitral valve prostheses type (*n*, %)	
Mechanical prosthesis	80 (83.3%)
Bioprosthesis	16 (16.7%)
Procedure indication (*n*, %)	
Heart failure (HF)	57 (59.4%)
Hemolytic anemia (HA)	5 (5.2%)
HF and HA	34 (35.4%)
Comorbidities (*n*, %)	
Previous CAD	21 (21.9%)
HTA	54 (56.3%)
DM	20 (20.8%)
AF	82 (85.4%)
Pulmonary hypertension (PASP > 60 mmHg by TTE)	52 (54.2%)
CKD (GF < 60 mL/min)	29 (30.2%)
COPD	9 (9.4%)
PAD	11 (11.5%)
NYHA Functional Class (*n*, %)	
I	3 (3.1%)
II	8 (8.3%)
III	68 (70.8)
IV	17 (17.7%)
Ejection fraction (%)	55.7 ± 9.4
Euroscore I	20.3 ± 10.1
STS score (mortality)	5.5 ± 4.0
Basal hemoglobin (mg/dL)	10.5 ± 1.8
Basal LDH (mg/dL)	775.5 ± 822.3
Previous HF admission (*n*, %)	40 (42.6%)
Red blood cells transfusion in last year (*n*, %)	34 (37.4%)

CKD: chronic kidney disease. COPD: chronic obstructive pulmonary disease. DM: diabetes mellitus. AF: Atrial fibrillation. HA: hemolytic anemia. HF: heart failure. HTA: arterial hypertension. PASP: pulmonary systolic pressure. PAD: peripheral artery disease.

**Table 2 jcm-11-04835-t002:** Procedural features.

Percutaneous Mitral PVL Procedures (*n*,)	128
PVL location (*n*, %)	
Lateral	42 (32.8%)
Anterior	37 (28.9%)
Septal	19 (14.8%)
Posterior	8 (6.3%)
Multiple	22 (17.2%)
Access (*n*, %)	
Femoral	125 (97.7%)
Transapical	3 (2.3%)
Approach (*n*, %)	
Antegrade	28 (64.8%)
Retrograde	39 (30.5%)
Combined	6 (4.7%)
Loop (*n*, %)	
Arterioarterial	3 (2.3%)
Venoarterial	111 (86.7%)
Nonloop	13 (10.32%)
Both	1 (0.78%)
Complications (*n*, %)	
None	103 (80.5%)
Vascular	12 (9.4%)
Device embolization	3 (2.3%)
Disc interference	6 (4.7%)
Cardiac surgery	2 (1.6%)
Other	2 (1.6%)
Death	0
Device (*n*, %)	
AVP-III	110 (90.2%)
PDA	1 (0.82%)
VSD	4 (3.3%)
AVP-IV	1 (0.82%)
Others	4 (3.3%)
Procedural timing (*n*, %)	
Elective procedure	120 (93.7%)
Procedure during acute HF admission	8 (6.3%)
Technical success (*n*, %)	115 (89.8)
Procedural success (*n*, %)	102 (79.2)

**Table 3 jcm-11-04835-t003:** Predictors of procedural success.

Factor	*p* Value	
	Univariate Analysis	Multivariate Analysis
NYHA I–II vs. III–IV	0.046	0.093 (NS)
HF vs. HA indication	0.002	0.054
Transfusion (yes/no)	0.000	0.085 (NS)
Location (multiple vs. no multiple PVL)	0.009	0.030
More than 2 previous surgeries	0.015	NS
Basal hemoglobin	0.063	NS
Basal LDH	0.000	NS
Pulmonary hypertension	0.135	-

HA: hemolytic anemia. HF: heart failure. PVL: paravalvular leak.

**Table 4 jcm-11-04835-t004:** Medium- and long-term outcomes.

Patients (*n*)	96
**90-Day Follow-Up**	
Improvement in NYHA functional class at 90 days (*n*, %)	75 (78.1%)
NYHA functional class at 90 days (*n*, %)	42 (32.8%)
I	37 (28.9%)
II	19 (14.8%)
III	8 (6.3%)
IV	22 (17.2%)
**Long-Term Follow-Up**	
Improvement in NYHA functional class at last follow-up visit (*n*, %)	74 (77.4%)
NYHA functional class at last follow-up visit (*n*, %)	
I	26 (27.4%)
II	53 (54.8%)
III	9 (9.5%)
IV	8 (8.3%)
Total MACEs at last follow-up visit (*n*, %)	55 (57.3%)
Death	38 (39.6%)
Cardiovascular death	65 (67.7%)
HF readmission	30 (31.3%)
Repeat PVL closure intervention	14 (14.6%)
Surgical repeat PVL intervention	11 (11.5%)
Percutaneous repeat PVL intervention	3 (3.1)
Stroke	0

**Table 5 jcm-11-04835-t005:** Predictors of major adverse cardiovascular events (MACE) during long-term follow-up.

	*p* Value	HR	95% Confidence Interval
CKD	0.043	2.1	1.0–4.6
3 months HF previous admission	0.000	5.7	2.3–13.8
First procedure success	0.002	0.1	0.0–0.5
Improve NYHA class at follow-up	0.011	0.4	0.4–0.8
HA vs. HF indication	0.000	5.9	2.9–12.2

CKD: chronic kidney disease. HA: hemolytic anemia. HF: heart failure.

**Table 6 jcm-11-04835-t006:** Predictors of death during long-term follow-up.

	*p* Value	HR	95% Confidence Interval
3 months HF previous admission	0.000	7.0	2.8–16.5
NYHA IV functional class 90 days after PVL closure	0.030	4.5	1.2–17.5
Improve NYHA class at follow-up	0.004	0.2	0.1–0.6
HA vs. HF indication	0.000	6.7	2.8–16.5

HA: hemolytic anemia. HF: heart failure.

## Data Availability

Not applicable.

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
