# Peer review of "Percutaneous Closure of Mitral Paravalvular Leak: Long-Term Results in a Single-Center Experience"

_jcm, 2022, doi:10.3390/jcm11164835_

Round 1
Reviewer 1 Report
In a paper by Gonzalez et al, the authors aimed to to evaluate the mid- and long-term FU of pts undergoing percutaneous closure of mitral PVL. During 10year study period, they were able to enroll 128 consecutive pts in whom 96 procedures were performed (73% were successful). It is as observational study yielding good long-term results of the PVL closure.
My remarks:
1. For PVL sizing, did the authors more rely on TEE or CT images or maybe intraprocedural angio or even balloon scoring? This aspect is also worth to be discussed later.
2. What about ethical issues? Please add it in the methods section.
3. Lines 129-134 should be place in the results section.
4. Study limitation section should be added commenting on relatively low sample size and reasons for that.
5. English language requires extensive editing. Please ask a native speaker for support.
Reviewer 2 Report
The overall work is good. There are some corrections and additive materials to make it even better:
1. Material and methods: please remove the sentence regarding animal
2. Analyzed variables: please remove the word previously on line 80
3. Missing reference in line 116
Please remove the paragraph starting on line 126 to the results section
Results
1. Please include numbers with only one decimal
20% complication rate is not low. Please correct it throughout the entire manuscript
Line 189: please correct the CI
Please replace the second paragraph of the results with the first one
Please show all variables that were used in the univariate analysis
What about ejection fraction, LA volume index, AF, etc.,
Table 1. Some words are not in English. Please mention below the units and acronyms used in the table
Include full names in the Kaplan-Meyer.
Please add an LDH curve and MACE (continuous)
Remove X2 from brackets
The discussion was too long. Focus on the adding value in your experience
Add central illustration
